Species variability of Pinus elliottii, P. elliottii×P. caribaea, and P. massoniana in response to pinewood nematode infection

Zeng Ming 1 2
Che Xiaoliang 1 2
Wang Zhe 1 2
Liu Yang 1 2
Deng Leping 3
Guo Yehuang 1 2
Huang Ting 3
Guo Wenbing wbguo@sinogaf.cn 1 2
1 Guangdong Academy of Forestry , Guangzhou , China
2 Guangdong Provincial Key Laboratory of Silviculture, Protection and Utilization , Guangzhou , China
3 Taishan Hongling Seed Garden , Taishan , China
Yapıcı Sercan
Electronic publication date: 2025 Sep 5
Publication date: 2025
Volume: 13
Electronic Location ID: e19990
Received 2025 Apr 16; Accepted 2025 Aug 5
Copyright: ©2025 Zeng et al.
Copyright year: 2025
Copyright holder: Zeng et al.
License: This is an open access article distributed under the terms of the Creative Commons Attribution License, which permits unrestricted use, distribution, reproduction and adaptation in any medium and for any purpose provided that it is properly attributed. For attribution, the original author(s), title, publication source (PeerJ) and either DOI or URL of the article must be cited.
License URL: https://creativecommons.org/licenses/by/4.0/

Keywords: Pinus species, Pine wilt disease, Bursaphelenchus xylophilus, Species variability, Artificial inoculation

Funding: Forestry Administration of Guangdong Province 2023KJCX018 Guangzhou Municipal Science and Technology Program 2024A04J4207 This work was supported by the Science and Technology Program from Forestry Administration of Guangdong Province (No. 2023KJCX018) and Guangzhou Municipal Science and Technology Program (No. 2024A04J4207). The funders had no role in study design, data collection and analysis, decision to publish, or preparation of the manuscript.

==============================
Pine wilt disease (PWD), caused by Bursaphelenchus xylophilus, poses a severe threat to global pine forests. This study evaluated the susceptibility of one-year-old seedlings of three pine species—Pinus massoniana, P. elliottii × P. caribaea, and P. elliottii—to PWD through artificial inoculation. Results showed that all species were generally susceptible at the seedling stage, developing symptoms and experiencing high mortality within a short period. However, P. elliottii, native to North America, exhibited the highest resistance, with slower disease progression and reduced pinewood nematode spread. P. elliottii × P. caribaea showed intermediate susceptibility, while the native P. massoniana was the most vulnerable, with rapid symptom onset and extensive xylem damage. These findings provide insights into species-specific resistance and inform breeding strategies for PWD management.

Introduction

Pine wilt disease (PWD), caused by the pinewood nematode (PWN; Bursaphelenchus xylophilus), poses an escalating threat to pine forests worldwide, inflicting significant ecological and economic damage. Endemic to North America, where its impact is generally mild on native pine species, PWD has rapidly spread across Asia and Europe over the past century (Futai, 2013; Inácio et al., 2015; Back et al., 2024; Robinet et al., 2024). China has experienced unprecedented destruction, with PWD wiping out millions of hectares of pine forests and incurring losses of over CNY 15 billion annually (Zhou et al., 2024). This expansion is primarily driven by Monochamus beetles, which act as vectors for B. xylophilus during their maturation feeding and during oviposition, introducing the nematode into pine trees through feeding wounds (Firmino et al., 2017). The nematodes, carried in a dispersal form within the tracheae of the beetles, can infect new trees as the beetles migrate. Different species of Monochamus, including M. alternatus in East Asia and M. galloprovincialis in Europe, play critical roles in the disease’s spread across large areas (Akbulut & Stamps, 2012; Vicente et al., 2012). Global trade and the movement of untreated wood products further facilitate the spread of B. xylophilus, as the nematode can be transported in timber and wood packaging materials. Climate change has also exacerbated the issue by creating more favorable conditions for both the nematode and its insect vectors, leading to increased disease outbreaks (Robinet et al., 2024; Zhou et al., 2024).

B. xylophilus has a complex life cycle that includes both phytophagous and mycophagous phases, allowing it to thrive in diverse environments and host conditions. In the phytophagous phase, the PWN feed on the cells within the resin ducts, leading to rapid wilting and eventual death of the tree. In the mycophagous phase, the PWN feeds on fungi present in dying or dead trees, enabling it to survive even after the host tree is no longer viable (Mamiya, 1983; Pimentel, Firmino & Ayres, 2021). These feeding strategies, coupled with the nematode’s rapid reproductive cycle, allow B. xylophilus to spread aggressively, causing widespread damage to pine forests. Upon entering the tree, B. xylophilus interferes with the tree’s vascular system, halting the flow of resin, which is typically one of the earliest signs of nematode infection (Jones et al., 2008). The nematodes trigger the formation of xylem embolisms—air blockages within the xylem vessels—that further disrupt water transport throughout the tree. This xylem dysfunction exacerbates water stress, leading to rapid wilting of needles, which turn yellow before browning and eventually dropping off (Futai, 2013; Gao et al., 2017). The embolisms spread from the initial infection site in a process known as “runaway embolism,” where large sections of xylem experience sudden and severe blockage, causing a sharp decline in water potential and accelerating the tree’s wilting process. These symptoms can manifest within weeks to months, and tree death often follows shortly thereafter due to the compounded effects of resin flow disruption and extensive xylem blockage (Umebayashi et al., 2011; Zhou et al., 2024).

Susceptibility to PWD varies significantly among different Pinus species, with some showing high vulnerability while others exhibit strong resistance to infection by B. xylophilus. In East Asia, species such as P. massoniana and P. thunbergii are notably susceptible, succumbing to nematode infestation (Toda & Kurinobu, 2002; Guo et al., 2023). Conversely, North American species like P. taeda demonstrate greater resistance, as do Mediterranean species such as P. pinea (Rodrigues et al., 2017; Menéndez-Gutiérrez et al., 2018a). The exact mechanisms responsible for these differences in susceptibility remain unclear. However, studies suggest that variations in nematode migration and reproductive capacity within different host species may contribute to these resistance patterns. For instance, pinewood nematodes have shown higher reproductive and migratory abilities in P. thunbergii compared to P. taeda, which could partially explain the differences in susceptibility observed between these species (Feng et al., 2022). In addition, transcriptomic analyses suggest that more resistant species may activate defense-related genes earlier and more selectively than susceptible ones. Studies of P. pinaster (susceptible) and P. yunnanensis (potentially more resistant) indicate that P. yunnanensis might upregulate genes involved in oxidative stress responses and the biosynthesis of defensive compounds, such as phenolics, even at early stages of infection (Gaspar et al., 2020). This timely and focused gene activation could help contain nematode spread more effectively in P. yunnanensis compared to the broader response observed in the more susceptible P. pinaster.

In southern China, Pinus elliottii, P. elliottii × P. caribaea hybrids, and P. massoniana are three important afforestation species, widely planted for timber and resin production (Guo et al., 2023; Dai et al., 2024). Among them, P. massoniana suffers the highest rates of infection by the PWN, followed by the P. elliottii× P. caribaea hybrid, while P. elliottii shows the strongest resistance (W. Guo, pers. obs., 2021–2023). These differences may be related to their origins. P. elliottii, a North American species native to the southeastern United States, is likely to have resistance potential against PWN. The hybrid pine (P. elliottii×P. caribaea), a cross between P. elliottii and P. caribaea (a species native to Central America and the northern West Indies), has been developed through hybrid breeding in China for over 30 years. In contrast, P. massoniana, a native species in China, may have a lower frequency of resistant germplasm in both natural populations and artificially selected populations. Previous studies have confirmed the existence of resistant germplasm in P. massoniana (Liu et al., 2020; An et al., 2023; Guo et al., 2023; Xie et al., 2023), demonstrating the potential for breeding P. massoniana germplasm with enhanced resistance. However, there is still a lack of comparative studies on the resistance responses of P. massoniana, P. elliottii, and P. elliottii×P. caribaea, as well as mechanistic insights into their differences. Understanding these mechanisms could facilitate genetic improvement and the development of P. massoniana germplasm with higher resistance. In this study, seedlings of P. elliottii, P. elliottii × P. caribaea, and P. massoniana sourced from seed orchards in Guangdong Province were artificially inoculated to investigate whether resistance is associated with controlling nematode migration and reproduction within the tree or tolerance to nematode presence even at higher densities. These findings aim to provide insights into the development of resistant germplasm and sustainable management strategies for PWD.

Materials and Methods

Plant material

The study was conducted in a nursery located in Tianhe District, Guangzhou City (113°37′E, 23°18′N). The site is situated at an elevation of 25 m above sea level and experiences a typical subtropical monsoon climate. The average annual temperature is about 22 °C, with January and July being the coldest and hottest month. The region receives an average annual rainfall of approximately 1,800 mm, with the majority occurring during the summer months. The inoculations were carried out in greenhouse conditions (average temperature of 26° ±2 °C, 60–80% humidity) at the nursery.

This study involved one-year-old seedlings (100 individuals of each species) of P. elliottii (72.5 ± 0.6 cm height, 8.05 ± 0.12 mm ground diameter), P. elliottii × P. caribaea hybrids (87.8 ± 1.1 cm height, 9.60 ± 0.21 mm ground diameter), and P. massoniana (50.3 ± 1.1 cm height, 9.60 ± 0.21 mm ground diameter). The pine seedlings used in this study were nursery-grown from seeds collected from seed orchards of the three tested species in Guangdong Province. Once germinated and established, the seedlings were transplanted to an isolated nursery for further growth. Two weeks before inoculation, the seedlings were moved into the greenhouse. The plants were watered twice every day, ensuring that the soil remained adequately moist but avoiding waterlogging. Watering was conducted at a consistent rate across all treatment groups to maintain uniform soil moisture conditions.

Culture of pinewood nematode

The B. xylophilus isolate was a virulent PWN cultured by the South China Agricultural University. The nematodes were maintained and propagated on a fungal culture of Botrytis cinerea, grown on potato dextrose agar (PDA) in Petri dishes under laboratory conditions.

To establish nematode cultures, actively feeding nematodes were transferred to the fungal colonies using a sterilized fine needle. The plates were incubated at 25 ± 1 °C for 7–10 days, during which the nematodes proliferated. After sufficient nematode growth, the nematodes were extracted using the Baermann funnel technique. Extracted nematodes were rinsed with sterile distilled water, quantified under a stereomicroscope, and used for subsequent inoculations.

Inoculation method and experimental setup

We used a serrated-incision inoculation method adapted from previous studies (Togashi et al., 1997; Carrasquinho et al., 2018; Menéndez-Gutiérrez et al., 2018b), making a cut 20 cm above the ground to mimic Monochamus-beetle wounding and facilitate nematode entry (Fig. 1A). Unlike the original protocols, our streamlined approach omits wound-moisture treatments and reduces handling time while still ensuring consistent nematode delivery and effective migration into the plant. The nematode suspension (100 µL) was applied directly to the exposed xylem. Three inoculation concentrations were used: 2,000 nematodes per tree, 4,000 nematodes per tree, and 8,000 nematodes per tree. A control group was treated with distilled water. Each treatment for each tree species included 24 replicates.

Figure 1 Experimental inoculation method and representative symptoms following nematode infection in different pine species.

(A) Schematic representation of pinewood nematode (B. xylophilus) inoculation procedure in pine seedlings, showing the incision method used for inoculation. (B) Representative symptoms observed in P. massoniana seedlings. (C) Representative symptoms observed in P. elliottii × P. caribaea hybrid seedlings. (D) Representative symptoms observed in P. elliottii seedlings. In (B), (C), and (D), the five columns from left to right represent seedlings categorized under the disease severity index, assigned on a scale from 0 to 4, where 0 indicates no symptoms, and 4 represents severe wilting and chlorosis affecting more than 75% of the tree.

Disease observation and severity index

After inoculation, the disease symptoms were recorded weekly, and each tree was assigned a disease severity index (0 to 4), where 0 indicated no symptoms, and 4 indicated severe wilting and chlorosis of more than 75% of the tree. The susceptibility rate (Rempel & Hall, 1996; Qiu et al., 2023) and mortality rate were calculated for each treatment group using the following formulas: Susceptibility rate=number of symptomatic trees/total number of trees×100%

Mortality rate=number of dead trees/total number of trees×100%.

Stem section observation and nematode migration analysis

Based on the results of preliminary experiments, B. xylophilus infection was found to cause structural damage throughout the entire stem of the seedlings. To ensure consistency in sampling and facilitate comparative analysis across different treatments, the position five cm above the base of the stem was selected for cross-sectional microscopy. Stem samples were collected at 35 days post inoculation from the stem of each plant, precisely five cm above the base. Cross-sectional observations were conducted using a DSX100 ultra-depth three-dimensional microscope (Olympus, Tokyo, Japan) to assess anatomical differences in the xylem and associated tissues. Structural variations between xylem tissues in plants with four grades of disease severity and those from the control group were systematically compared. Observations focused on the disruption of xylem integrity, resin canals, and other tissue alterations caused by nematode infection.

To evaluate nematode migration and reproduction within the plant, wood segments were collected above and below the inoculation site for the symptomatic trees also at 35 days post inoculation. A 10-cm segment was excised in both directions after removing the wound area. The fresh weight of each segment was measured, and the samples were finely chipped. Nematodes were extracted from the wood chips using the Baermann funnel method, with the setup maintained for 24 h to allow nematode migration into the extraction solution. The nematode-containing solution was collected and centrifuged at 5,000 rpm for 10 min. After removing the supernatant, the nematode pellet was resuspended in two mL of distilled water to prepare the identification solution. Three 50 µL subsamples from the identification solution were prepared on microscope slides for nematode counting under a stereomicroscope. Three biological replicates were included for each treatment, and counts were averaged across the subsamples. The density of nematodes within each wood segment was calculated using the formula: Nematode Density/gwood=Nematode count on slides×Total identification solution volume/50/Fresh weight of wood.

Statistical analysis

Statistical analyses were performed using the Statistical Product and Service Solutions (SPSS) 21.0 software. Data are expressed as the mean ± standard error (SE). Variance among multiple groups was compared using one-way analysis of variance (ANOVA), followed by Tukey’s multiple comparisons post-test. Statistical significance was considered at P < 0.05. Graphical representations of the data were generated using Microsoft Excel or OriginPro 8.0.

Results

Symptom development across tree species

Following inoculation with B. xylophilus, initial symptoms began to appear approximately one week post-inoculation, progressively intensifying over time. The severity of symptoms varied among species, with clear differences in disease progression and symptom expression (Figs. 1B–1D). Throughout the observation period, individuals within each species exhibited varying degrees of susceptibility, as indicated by the presence of trees across all disease severity index (DSI) levels.

At DSI 0, all species maintained healthy needle color and structure, showing no visible symptoms of infection (Figs. 1B–1D, first column). By DSI 1, P. massoniana exhibited mild chlorosis and noticeable needle drooping, particularly on lower branches (Fig. 1B, second column). In contrast, P. elliottii × P. caribaea showed yellowing predominantly in the upper canopy, with slight drooping in some branches (Fig. 1C, second column). P. elliottii, however, mainly exhibited needle yellowing without significant drooping, maintaining an upright structure (Fig. 1D, second column).

As infection progressed to DSI 2, symptoms intensified across all species. P. massoniana displayed widespread needle yellowing and increased drooping, particularly in the lower branches (Fig. 1B, third column). P. elliottii × P. caribaea showed a more symmetrical pattern of yellowing, with partial drooping but a relatively intact canopy structure (Fig. 1C, third column). In P. elliottii, needle yellowing became more pronounced, yet the tree maintained its structural integrity, with minimal branch drooping (Fig. 1D, third column).

At DSI 3, P. massoniana exhibited extensive needle browning, severe wilting, and overall plant collapse, with many branches losing their turgidity (Fig. 1B, fourth column). P. elliottii × P. caribaea also showed severe chlorosis and wilting, but some green foliage remained in the upper canopy (Fig. 1C, fourth column). Meanwhile, P. elliottii continued to display dominant yellowing symptoms rather than extensive drooping, differentiating it from the other two species (Fig. 1D, fourth column).

By DSI 4, P. massoniana was the most severely affected, with extreme wilting, extensive chlorosis, and complete needle loss, often leading to plant death (Fig. 1B, fifth column). P. elliottii × P. caribaea exhibited similar mortality symptoms, with its structure collapsing but some branches retaining discolored needles (Fig. 1C, fifth column). In contrast, P. elliottii, while exhibiting full chlorosis and needle desiccation, maintained a more upright structure, suggesting a different mode of symptom expression compared to the other two species (Fig. 1D, fifth column).

These results indicate that while all species were affected by B. xylophilus infection, P. massoniana was the most susceptible, with rapid symptom development and extensive wilting. P. elliottii × P. caribaea showed moderate susceptibility, exhibiting chlorosis and partial drooping, while P. elliottii demonstrated the highest resistance, primarily displaying chlorosis with minimal needle drooping.

Changes in xylem and pith structures

The effects of B. xylophilus infection on the xylem and pith were assessed by examining stem cross-sections from both healthy and infected plants (Fig. 2). In healthy plants (DSI 0), the xylem and pith were well-organized and structurally intact across all species (Figs. 2A–2C). The xylem exhibited clearly defined tracheid cells, and the pith appeared uniform, without signs of damage or abnormality. The resin canals (RC) were clearly visible, and the pith (P), located centrally within the stem, remained structurally undisturbed.

Figure 2 Cross-sectional observations of stem tissues in different pine species across disease severity index levels (DSI).

Representative stem cross-sections of P. massoniana (PM, left column), P. elliottii × P. caribaea hybrid (EH, middle column), and P. elliottii (PEE, right column) at different DSI levels. The DSI categories range from 0 (no visible symptoms) to 4 (severe tissue damage), corresponding to the disease severity classification used in Figs. 1 and 4. (A–C) DSI 0: healthy xylem and pith structures with no visible signs of infection. (D–F) DSI 1: slight discoloration in the xylem with minor structural changes. (G–I) DSI 2: moderate tissue degradation, including the formation of brown lesions. (J–L) DSI 3: extensive xylem disruption, increased resin accumulation, and visible necrosis. (M–O) DSI 4: severe tissue decay, extensive necrosis, and structural collapse of xylem and pith. RC, resin canal; P, pith. Scale bars: 500 µm (A, G); 1,000 µm (B–F, H–O).

As infection progressed, increasing damage to xylem and pith tissues was observed, with species-specific variations in severity (Figs. 2D–2O). At DSI 1, P. massoniana exhibited slight discoloration in the xylem and early-stage resin accumulation (Fig. 2D), while P. elliottii × P. caribaea showed minimal tissue changes with slight yellowing near the pith (Fig. 2E). P. elliottii displayed only minor resin deposition around the pith, with no evident structural collapse (Fig. 2F).

By DSI 2, P. massoniana exhibited moderate tissue degradation, characterized by brown lesion formation and increased resin deposition around the xylem (Fig. 2G). P. elliottii × P. caribaea showed a distinct necrotic patch in the pith, indicating progressive damage (Fig. 2H), while P. elliottii displayed clear necrotic changes within the pith but maintained overall xylem integrity (Fig. 2I).

At DSI 3, P. massoniana demonstrated extensive xylem disruption, resin accumulation, and visible necrotic patches throughout the vascular tissue (Fig. 2J). P. elliottii × P. caribaea exhibited severe necrosis, with resin canals appearing blocked and structural collapse evident in the pith (Fig. 2K). P. elliottii, although showing localized necrosis, maintained relatively intact xylem structure compared to the other two species (Fig. 2L).

By DSI 4, P. massoniana displayed complete vascular collapse, widespread necrosis, and severe structural degradation (Fig. 2M). P. elliottii × P. caribaea showed near-total loss of vascular integrity, with extensive resin accumulation and pith decay (Fig. 2N). In contrast, P. elliottii exhibited severe necrosis but retained partial xylem structure, suggesting a comparatively slower rate of tissue degradation (Fig. 2O).

These findings indicate that the severity of vascular tissue damage correlates with species susceptibility, with P. massoniana exhibiting the most pronounced xylem and pith disruption, P. elliottii × P. caribaea showing intermediate damage, and P. elliottii maintaining better structural integrity, consistent with its higher resistance to B. xylophilus.

Susceptibility rate, disease severity index, and mortality rates

The progression of pine wilt disease (PWD) in the three pine species was evaluated based on susceptibility rate, disease severity index (DSI), and mortality rate over a 35-day period following inoculation with B. xylophilus. The effects of different inoculation doses were also analyzed to determine the influence of nematode density on disease development.

The susceptibility rate, defined as the percentage of symptomatic plants, increased progressively in all species throughout the observation period (Fig. 3). At lower inoculation doses (2,000 PWN per plant, Fig. 3A), P. massoniana exhibited a rapid increase in susceptibility, reaching 80% by 21 days post-inoculation (dpi), while P. elliottii × P. caribaea and P. elliottii showed a slower progression, with final susceptibility rates of approximately 90% and 100%, respectively, by 35 dpi. At higher inoculation doses (4,000 and 8,000 PWN per plant, Figs. 3B and 3C), all species showed a more rapid increase in susceptibility, with nearly all plants developing symptoms by 28 dpi. These trends across all inoculation doses indicated that P. massoniana was the most susceptible early in the infection process, but by the later stages (28–35 dpi), the susceptibility rates among the three species were comparable.

Figure 3 Cumulative susceptibility rates of different pine species inoculated with B. xylophilus.

Cumulative susceptibility rates of different pine species inoculated with B. xylophilus at doses of 2,000 (A), 4,000 (B), and 8,000 (C) nematodes per seedling at each time point. Bars represent different tree species: PM (P. massoniana), EH (P. elliottii× P. caribaea), and PEE (P. elliottii).

Figure 4 Disease severity index distribution over time for different pine species under varying inoculation doses of B. xylophilus.

Stacked bar plots show the progression of disease severity index (DSI) in P. massoniana (PM, left column), P. elliottii × P. caribaea (EH, middle column), and P. elliottii (PEE, right column) at 7, 14, 21, 28, and 35 days post-inoculation. Different inoculation doses were applied: 2,000 PWN (A–C), 4,000 PWN (D–F), and 8,000 PWN (G–I) per seedling.

The progression of disease severity varied among species, with clear distinction observed in the distribution of plants across DSI categories (Fig. 4). At 7 dpi, the majority of plants remained in DSI 0 or 1, indicating minimal symptom development across all species. However, by 14 dpi, P. massoniana exhibited a substantial shift towards higher DSI levels, with many individuals reaching DSI 2 or 3, while P. elliottii × P. caribaea and P. elliottii retained a higher proportion of plants at lower DSI levels. By 21 dpi, P. massoniana had the highest proportion of plants in DSI 3 and 4, indicating severe wilting and chlorosis.

By 28–35 dpi, plants from all three species exhibited significant disease progression, but P. elliottii maintained a higher proportion of plants at DSI 2 or lower compared to the other two species, which had shifted predominantly to DSI 3 and 4. These results suggest that P. massoniana was the most susceptible to severe disease progression, followed by P. elliottii × P. caribaea, while P. elliottii exhibited the slowest disease development and retained a greater number of less severely affected individuals.

Mortality rates followed a similar trend to the DSI progression, with increasing mortality observed over time and at higher inoculation doses (Fig. 5). At 2,000 PWN per plant (Fig. 5A), P. massoniana exhibited the highest mortality rate by 35 dpi, while P. elliottii had the lowest. Increasing the inoculation dose to 4,000 and 8,000 PWN per plant (Figs. 5B and 5C) resulted in significantly higher mortality across all species, with nearly 100% mortality observed in P. massoniana and P. elliottii ×P. caribaea at 35 dpi.

Figure 5 Cumulative mortality rates of different pine species inoculated with B. xylophilus.

Cumulative mortality rates of different pine species inoculated with B. xylophilus at doses of 2,000 (A), 4,000 (B), and 8,000 (C) nematodes per seedling at each time point. Bars represent different tree species: PM (P. massoniana), EH (P. elliottii × P. caribaea), and PEE (P. elliottii).

Across all tested inoculation doses, P. massoniana consistently showed the highest mean susceptibility rate, disease severity index, and mortality rate. P. elliottii ×P. caribaea exhibited an intermediate response, while P. elliottii demonstrated greater resistance, showing slower symptom development, lower DSI scores, and reduced mortality, even at high inoculation doses. These results highlight the potential for utilizing P. elliottii in breeding programs aimed at enhancing resistance to PWD in pine plantations.

Nematode migration within the plant

To compare the migration and reproductive efficiency of B. xylophilus across different Pinus species, PWN density was assessed at the upper and lower sections of the inoculation site at various inoculation doses (Table 1; Fig. 6). The analysis revealed significant differences in PWN density among tree species (P < 0.05) and inoculation doses (P <  0.05), but no significant difference in the direction of PWN migration (P > 0.05), indicating that PWNs migrated equally in both upward and downward directions across all species and doses.

Table 1 Analysis of variance (ANOVA) for the effect of species, direction (B. xylophilus migration direction), dose (inoculation dose), and their interactions on B. xylophilus density.

Source of variation	DF	F value	P value	
Species	2	9.19	0.0005	
Direction	1	0.94	0.3379	
Dose	2	194.02	<0.0001	
Species ×dose	4	1.27	0.2974	

Figure 6 Comparison of Bursaphelenchus xylophilus density across different Pinus species and inoculation doses.

(A) Nematode density among tree species (PM: P. massoniana , EH: P. elliottii × P. caribaea , PEE: P. elliottii) with different inoculation doses (2,000, 4,000, 8,000 nematodes per plant), combining both upper and lower portions. (B) Nematode density across different inoculation doses (2,000, 4,000, 8,000 nematodes per plant), combining both upper and lower portions, across all tree species. (C) Nematode density in the upper portion of the inoculation site. (D) Nematode density in the lower portion of the inoculation site. Bars represent mean±SE. Different lowercase letters indicate statistically significant differences (P < 0.05).

Regarding species differences, P. massoniana exhibited significantly higher PWN densities than P. elliottii × P. caribaea and P. elliottii (Fig. 6A). These results suggest that P. massoniana may be more favorable for PWN settlement compared to the other species, although the overall migration of PWNs into the stem was similar across species.

With respect to inoculation doses, PWN density increased significantly with higher inoculation doses (Fig. 6B). The highest PWN density was observed at the 8,000 PWNs per plant dose, followed by 4,000 and 2,000 doses. This dose-dependent increase in PWN density suggests that greater inoculation pressure leads to higher PWN migration and reproduction within the plant.

The interaction between tree species and inoculation dose was further explored by comparing PWN density in the upper and lower portions of the inoculation site (Figs. 6C, 6D). In the upper portion (Fig. 6C), significant differences in PWN density were observed at the 4,000 and 8,000 inoculation doses. P. massoniana exhibited the highest densities, followed by P. elliottii × P. caribaea and P. elliottii. Similarly, in the lower portion (Fig. 6D), significant differences were observed at the 4,000 and 8,000 doses, with P. massoniana showing the highest densities, followed by P. elliottii × P. caribaea and P. elliottii. These results indicate that PWNs migrate similarly in both directions across species, but species still exhibit varying levels of resistance at higher inoculation doses.

Discussion

This study provides a comprehensive analysis of the susceptibility of three Pinus species to B. xylophilus infection, with a focus on tissue damage, disease progression, and PWN migration. The results highlight significant differences in resistance and susceptibility among species, which may be attributed to both genetic factors and physiological traits. The study offers insights into the common tissue damage across species, and the role of nematode migration and reproduction in disease progression. Additionally, the inoculation method used in this study proved effective for all three tree species. This method offers several advantages, including its simplicity and efficiency in delivering B. xylophilus to the host plants. The method can be easily replicated in different research settings and is particularly useful for studying disease resistance across various tree species, making it an important tool for future investigations into the management of pine wilt disease.

A striking common feature across all three species was the irreversible damage to the xylem and pith following inoculation. Infected plants exhibited significant disruption of resin canals, a key characteristic of pine wilt disease (Mamiya, 1983; Futai, 2013). The destruction of these tissues led to a reduction or complete cessation of resin production, impairing the tree’s ability to defend itself against further nematode invasion. The disruption of water transport within the plant was also evident, as indicated by the wilting and desiccation of needles, which were unable to maintain proper photosynthesis. This ultimately resulted in the death of the entire plant. This sequence of events is consistent with previous studies describing the pathological process of pine wilt disease, where B. xylophilus impedes vital physiological functions, ultimately leading to tree mortality (Gao et al., 2017).

The disease progression followed a predictable pattern across all tree species, with symptoms of wilting and yellowing appearing within a few weeks post-inoculation. Notably, the susceptibility rate, disease severity index and mortality rates varied significantly among species. P. massoniana displayed the fastest disease progression, with higher susceptibility observed at both low and high inoculation doses. Conversely, P. elliottii exhibited slower disease progression and lower susceptibility, suggesting that this species may have a more effective immune response to nematode invasion. P. elliottii × P. caribaea exhibited intermediate resistance, confirming that hybrid species can sometimes exhibit a blend of the parental species’ resistance traits. These findings support earlier studies that have shown species-specific differences in disease progression, with southern pines such as P. elliottii being more resistant than native East Asian pines like P. massoniana (Futai, 2013; Rodrigues et al., 2017).

The rate of disease progression was also influenced by the inoculation dose. At higher doses, the disease progressed more rapidly, with higher mortality rates observed across all species. This underscores the role of nematode population density in determining the severity of the disease. The dose-dependent progression of disease symptoms has been documented in other studies and reinforces the importance of nematode density in determining the overall outcome of pine wilt disease (Zhou et al., 2024).

Nematode migration and reproductive efficiency are crucial factors that determine the success of the nematode in infecting and colonizing the host plant. For example, in tomato (Solanum lycopersicum) infected with the root-knot nematode Meloidogyne incognita, higher initial inoculum densities lead to proportionally greater gall formation and egg mass production, demonstrating a clear density-dependent increase in reproduction and thus disease severity (Kamran et al., 2013). Our study found that the initial inoculation dose significantly impacted nematode migration and/or reproduction within the plant. At higher inoculation doses (8,000 nematodes per plant), P. massoniana exhibited significantly higher nematode densities compared to P. elliottii. This suggests that the latter species (PEE) is more effective at restricting nematode movement and/or reproduction. The migration patterns of the nematodes also showed that nematodes tend to move both upward and downward from the initial inoculation site, with nematodes generally migrating and/or reproducing more within the susceptible species. These findings support the hypothesis that species with more efficient immune responses may restrict nematode reproduction, preventing further spread within the plant (Gao et al., 2017). Furthermore, field surveys in southern China have shown that P. elliottii exhibits lower incidence and slower progression of PWD under natural conditions compared to P. massoniana and the P. elliottii × P. caribaea hybrid (W Guo, pers. obs., 2021–2023). These observations align with our artificial-inoculation results and strongly support the use of inoculation tests on young seedlings in breeding programs.

Conclusion

Our study revealed that all three Pinus species tested—P. massoniana, P. elliottii × P. caribaea, and P. elliottii—were generally susceptible to Bursaphelenchus xylophilus at the seedling stage, showing symptoms and high mortality within a short period post-inoculation. However, relative differences in susceptibility were observed. P. elliottii, originating from North America, exhibited the highest resistance, with slower disease progression and lower nematode migration and/or reproduction efficiency. The hybrid P. elliottii × P. caribaea showed intermediate resistance, while the native P. massoniana was the most susceptible, with rapid symptom development and extensive xylem damage. These findings provide a basis for breeding strategies aimed at improving resistance to pine wilt disease.

Supplemental Information

Supplemental Information 1 Data of susceptibility rates

note: I, infected; U, uninfected

Supplemental Information 2 Data of Disease severity index

Each tree was assigned a disease severity index (0 to 4), where 0 indicated no symptoms, and 4 indicated severe wilting and chlorosis of more than 75% of the tree

Supplemental Information 3 Data of mortality rates

note: L, live; D, dead

Supplemental Information 4 Data of Bursaphelenchus xylophilus density

ND, non detectable

Additional Information and Declarations

Competing Interests

Author Contributions

Data Availability

The authors declare there are no competing interests.

Ming Zeng performed the experiments, analyzed the data, prepared figures and/or tables, and approved the final draft.

Xiaoliang Che performed the experiments, prepared figures and/or tables, and approved the final draft.

Zhe Wang analyzed the data, prepared figures and/or tables, and approved the final draft.

Yang Liu analyzed the data, authored or reviewed drafts of the article, and approved the final draft.

Leping Deng performed the experiments, authored or reviewed drafts of the article, and approved the final draft.

Yehuang Guo performed the experiments, prepared figures and/or tables, and approved the final draft.

Ting Huang performed the experiments, authored or reviewed drafts of the article, and approved the final draft.

Wenbing Guo conceived and designed the experiments, analyzed the data, authored or reviewed drafts of the article, and approved the final draft.

The following information was supplied regarding data availability:

The raw data is available in the Supplemental Files.

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
