# Peer review of "Species variability of Pinus elliottii, P. elliottii×P. caribaea, and P. massoniana in response to pinewood nematode infection"

_PeerJ, doi:10.7717/peerj.19990_

## Round 0.1 · original submission · Minor Revisions

Dear Dr. Guo,
You can find the comments and suggestions of the expert reviewers in the attached reports. As you will see, expert reviewers have pointed out the errors. Consequently, a minor revision is needed for your article.

I request that you improve your manuscript following the reviewers' suggestions
Sincerely

**PeerJ Staff Note**: Please ensure that all review, editorial, and staff comments are addressed in a response letter and that any edits or clarifications mentioned in the letter are also inserted into the revised manuscript where appropriate.

**PeerJ Staff Note**: It is PeerJ policy that additional references suggested during the peer-review process should only be included if the authors agree that they are relevant and useful.

Reviewer 1 ·

Basic reporting

The article is well written. I couldn't find any mistakes. English is good as well

Experimental design

-

Validity of the findings

-

Reviewer 2 ·

Basic reporting

Very clear and professional English is used throughout the manuscript. I would only suggest a minor modification of the title for more clarity by moving “in response”: “Species Variability of Pinus elliottii, P.elliottii × P. caribaea, and P. massoniana in response to Pinewood Nematode Infection”.

Experimental design

-

Validity of the findings

-

Additional comments

The manuscript entitled “Species Variability in response of Pinus elliottii, P.elliottii × P. caribaea, and P. massoniana to Pinewood Nematode Infection” by Zeng M. et al shows very rigorous and important work from experiment to results analysis, and clear and meaningful discussions. I would recommend the publication of this work with only minor modifications.

The introduction part clearly explains the context of the study based on relevant literature and leads to a good presentation of the research question. Figures and tables are relevant and well described. Raw data is supplied.

Methods are generally well described. About the inoculation method: it is simple and fast and slightly different from the one used in other publications (eg, Menendez-Gutierrez et al. doi:10.1093/forestry/cpx030, Carrasquinho et al https://doi.org/10.1007/s13595-018-0759-x, Togashi et al J. For. Res. 2:39-43 1997) since there is no protection of the wound to avoid desiccation. In the discussion part (line 296), the authors stress that, based on their results, this simple method is efficient and could be helpful in other studies. It would then be interesting in the Methods part to better explain what it is original (or adapted from other studies) and expose its advantages compared with other methods.

Results are very clearly presented and analysed. I would suggest minor additions. On susceptibility rate analysis (line 235 and Figure 3), there is no mention of significativity of observed species differences, which is expected since there is no repetitions for each species at each date at each dose, but when combining all inoculum doses (Figure 3D), an anova analysis could be performed at each date to test species differences. The same applies to the mortality rate (line 255 and Figure 5D). Line 244, it is said that “The progression of disease severity varied among species, with significant differences observed in the distribution of plants across DSI categories (Figure 4).”It is not clear from which ANOVA model those significant differences were observed; the authors should reformulate this part. Line 264, it is said that “The combined analysis of susceptibility rate, disease severity index, and mortality rate indicates that P. massoniana is the most susceptible species to PWD”, which is a result well supported by all presented data. However, “combined analysis” usually refers to an overall analysis of repeated experiments to estimate an average response and the consistency of the responses, which is what the authors did by testing species responses with different PWN inoculum doses. Then the above sentence could be “Across all tested inoculation doses, the combined analysis …”, and authors could add whether this effect was significant or not after adding an anova analysis as said above (Figure 3D and 5D).

The discussion part is well structured and based on relevant references. However, I would recommend more caution when discussing nematode migration and reproductive efficiency factors (line 324 and after) since those factors are confounded when measuring nematode density in the plant stem. For example, it is said “Our study found that the initial inoculation dose significantly impacted both nematode migration and reproduction within the plant.”, but in my understanding there is no evidence that both factors were impacted since nematode density was measured in only two stem sections (above and below wound) and there was no difference in density between those sections across species or doses. I think there is evidence that the nematode reproductive efficiency is different between species (if stem biomasses are similar between species but nematode densities are different), but there is no evidence about migration: it could or could not be affected. To demonstrate a migration difference between species or inoculum doses, one would expect a comparison of nematode densities at different distances from the wound. In the absence of such results, I would rather say the “study found that the initial inoculation dose significantly impacted nematode migration and/or reproduction within the plant,” and lower. “This suggests that the latter species (PEE) is more effective at restricting nematode movement and/or reproduction.” Also, there is no evidence in this specific study supporting the statements as written, lines 330: “with nematodes generally migrating more extensively within the susceptible species” or line 339: “lower nematode migration ».

In the Conclusion (or discussion) part, it could be interesting to draw a parallel between the findings of this study on juvenile seedlings with artificial inoculation and the observed susceptibility of the same species stands in epidemic conditions in China (as said lines 84-86 in the introduction): this would strongly support the use of inoculation test on young seedlings in breeding programs.

Last comment (anecdotal): lines 207-208, “The resin canals (RC) … and the pith (P)”: I couldn’t find the use of those abbreviations (not seen in Figure 2).

·

Basic reporting

1. Regarding the Introduction, there is a significant amount of general description. The descriptions from L28 to L81 should be limited to the situation of pine wilt disease in China.

2. I believe that “overall” in Figures 3 and 5 is unnecessary. This is because it is important to understand the effect of differences in inoculation density on symptoms, and I believe it is more important to clearly show the results for each case.

Experimental design

I believe that the experiment was conducted appropriately.

Validity of the findings

I understand this study to be a report that clarifies the differences in resistance among pine species to pine wilt disease. I consider the results to be clear.
Indeed, I believe there are no papers that directly compare the resistance of Pinus massoniana, P. elliottii × P. caribaea, and P. elliottii. In that sense, it is novel. Additionally, I believe there have been no papers in the past ten years that focused solely on how changes in inoculation density affect the proportion and rate of mortality. Therefore, this is also novel.

In addition to the fact that novelty and impact are not subject to review by Peer J, no papers consistent with this paper have been published to date. Considering these two points, I have decided that I have no choice but to agree to the publication of this paper.

Additional comments

Regarding the Discussion, the descriptions of population density and symptoms should include examples from other plants. This is because this section is the only part of the paper that I consider to have unique value.

---

## Round 0.2 · accepted · Accept

Dear Dr. Guo,

I thank you for making the corrections and changes requested by the reviewers. I read and checked your valuable article carefully and am happy to inform you that the article has been accepted for publication in PeerJ.

Sincerely yours,

Reviewer 2 ·

Basic reporting

no comment

Experimental design

no comment

Validity of the findings

no comment

Additional comments

The authors have carefully and thoughtfully addressed all comments and suggestions of the reviewers in this revised version of thieir manuscript. It is now fully suitable for publication in your journal.

·

Basic reporting

no comment

Experimental design

no comment

Validity of the findings

no comment

Additional comments

no comment